# Receptivity and Remating Propensity in Female *Spodoptera litura* (Fabricius) after Mating with an Irradiated Male or Its F_1_ Male Progeny

**DOI:** 10.3390/insects14070651

**Published:** 2023-07-20

**Authors:** Nilza Angmo, Madhumita Sengupta, Neha Vimal, Rakesh Kumar Seth

**Affiliations:** Applied Entomology and Radiation Biology Lab, Department of Zoology, University of Delhi, Delhi 110007, India; nilzalampa@gmail.com (N.A.); madhumitasg92@gmail.com (M.S.); nehavimal220@gmail.com (N.V.)

**Keywords:** insemination quality, female receptivity, calling behavior, inter-mating interval

## Abstract

**Simple Summary:**

*Spodoptera litura* (Fabr.), an important pest of agricultural crops in India, can be controlled using the ‘Inherited or F_1_ sterility technique’. The multiple mating tendency in female moths, their receptivity to remating after initial mating with radio-sterilized males and the sperm use pattern in sequential matings might influence the level of sterility induced in a radiation-mediated F_1_ sterility program. In this study, the effect of irradiation was assessed on the quality of the ejaculate of exposed male moths, with a substantial decrease in insemination quality in their F_1_ male progeny. Females mated to F_1_ males showed an increased calling efficiency and remating propensity (with normal males) and influenced reproductive performance with a prolonged post (initial)-mating interval compared with the control moths. These findings on factors influencing female receptivity during remating and eventual reproductive sterilization might help optimize the technique.

**Abstract:**

The ‘Inherited or F_1_ sterility technique’ (IS), using sub-sterilized male moths, is a widely proposed pest management tool for Lepidoptera pests in general, and the tobacco cutworm *Spodoptera litura* (Fabr.) in particular. However, the multiple mating tendency of female moths and the ejaculate quality of male moths might influence the efficiency of this technique. Reduced ejaculate quality was observed in irradiated males, as evidenced by radiation’s impact on certain bio-parameters, such as the weight of the spermatophores and their protein content, sperm count, the molecular expression of the sex peptide receptor (*SPR*) and egg fertility, with a greater impact in F_1_ male progeny. During the remating of females with untreated males, irrespective of the irradiation status of the first male, there was an increase in calling behavior, remating propensity and fertility in females, with a larger time gap between consecutive matings. The ability of F_1_ male progeny to check remating propensity in females 24 h after the initial mating was lower than that of unirradiated males. Partially sterile (130 Gy) males were as successful as unirradiated males in inducing the level of mating refractoriness in females. Decreased ejaculate quality in F_1_ male progeny could be associated with increased female receptivity during remating. Understanding the influence of male moth irradiation, insemination quality and post (initial)-mating intervals on the remating behavior of normal female moths and induced sterility might help in simulation modeling and optimizing IS insect programs.

## 1. Introduction

The tobacco cutworm *Spodoptera litura* (Fabricius) (Lepidoptera: Noctuidae) is an important agricultural and forest pest species [1], found in the tropical and temperate regions of Asia, Australia, Europe and the Pacific islands. It is a highly polyphagous and economically important pest with a wide host range and high reproductive fitness [2]. Chemical control used to be the primary management strategy against *S. litura* [3,4], but the development of resistance towards these chemicals and their detrimental effect on the environment led to the pursuit of alternate biological control methods [5,6]. An autocidal control method, commonly referred to as the sterile insect technique (SIT), has been used successfully for managing insect pest populations by manipulating their reproductive potential. It is an environmentally friendly control tactic that involves releasing sterilized mass-reared males into natural populations to mate with wild females [7]. In Lepidoptera, a modification of the SIT is used, known as the ‘Inherited or F_1_ sterility technique’ (IS), in which mass-reared and released males receive a sub-sterilizing dose of radiation before mating with wild females of the pest species, which then produce sterile F_1_ offspring [8].

In SIT and IS release programs, irradiated males are released in the target area, and they seek out wild females for mating [7]. In some dipteran insect pest species such as tsetse flies (Glossinidae) [9] and New World screwworm flies (Calliphoridae) [10], which have been managed via the SIT, wild females tend to mate only once. In Lepidoptera, females have an inherent tendency to mate multiple times [11,12,13,14,15], and this multiple mating behavior could be influenced by the initial mating with radio-sterilized males or their F_1_ male offspring [16]. A wild female that has mated with an irradiated released male could mate with an irradiated male, an F_1_ irradiated male or a wild unirradiated male during a second mating opportunity. According to Parker (1970), multiple matings by polyandrous females results in the presence of sperm from many males in the female spermatheca, which causes postcopulatory sperm competition among the males [17]. Therefore, the remating propensity and the factors influencing this behavior could have important implications for the outcomes of these SIT or IS programs.

After mating, the receptivity of the female moth is usually reduced for a certain period, but if its receptivity for mating is restored, then the female’s response towards unirradiated or irradiated males should ideally be similar. However, the propensity of female moths to remate might influence the effectiveness and efficiency of the SIT or IS technique [18,19] as later matings of the female with untreated wild males might displace, mix or incapacitate the first mate’s sperm [20]. The precedence of the sperm of the last mating has been reported in *S. litura* [21,22]. Hence, the reduced remating propensity of wild females post mating with irradiated males would contribute to the competitiveness of the released irradiated males.

The level of polyandry in moths can be influenced by the population density and composition (age and sex ratio) as well as climatic factors [23,24]. The multiple mating frequencies in female moths are related to their age at first mating [16] as well as male mating status [13]. In *S. littoralis,* females paired with mated males showed a higher remating propensity than those mated with virgin males [13]. Therefore, understanding the effect of the mating history of normal and irradiated males on their behavioral and reproductive performances, along with their ability to induce refractoriness in females, is equally important in the optimization of the IS technique. Recently, male mating history and irradiation’s effects on the ejaculatory quality and reproductive behavior of male moths of *S. litura* were studied. Multiple mating of irradiated male moths was found to lower the amount of sperm transferred, and it also affected copulation duration and mating success [25].

In many female insects, mating causes a temporary or permanent loss of sexual receptivity or a decrease in sexual attractiveness, and this has been associated with reduced calling behavior and/or the cessation of pheromone synthesis (pheromonostasis) in Lepidoptera [26,27,28]. Multiple factors have been associated with regulating the receptivity to mating in female moths, and these are often influenced by the quality and persistence of courting males. The factors resulting in the suppression of receptivity in mated females are derived from males. Some of these factors have a transient or short-term effect on the behavior of female moths, whereas others induce a long-term effect. Female refractoriness after mating might be a result of a combination of multiple factors, such as mechanical stimulation by the spermatophore during mating [29,30], secretions from the male accessory gland (MAG) (such as the sex peptide Acp70A in *Drosophila*) [26,31] or the presence of sperm in the spermatheca [32,33,34]. In insects such as the gypsy moths, *Lymantria dispar* Linnaeus (Lepidoptera: Erebidae), and thesilk moths, *Bombyx mori* Linnaeus (Lepidoptera: Bombycidae), long-term suppression of female receptivity is associated with the successful storage of sperm in their spermathecae [33,35]. Similarly, in the oriental armyworm, *Pseudaletia separata* Walker (Lepidoptera: Noctuidae), and the cotton bollworm, *Heliothis zea* Boddie (Lepidoptera: Noctuidae), the apyrene sperm count was found to be associated with female receptivity [36,37]. In many species, the spermatophore size could be linked to female receptivity, with larger spermatophores inducing longer refractory periods in mated females [38,39].

The male seminal fluid that is primarily produced by the accessory glands, but also by the testes or ejaculatory duct, has also been associated with suppressing female receptivity, in addition to stimulating egg maturation and oviposition [26,28,40]. The seminal fluid is transferred to the female’s reproductive tract, and the factors (such as sex peptide, etc.) present in the seminal fluid influence the nervous system via the hemolymph [41]. The sex peptide (SP) present in the seminal fluid acts as a trigger in *Drosophila melanogaster* (Diptera: Drosophilidae) for post-mating onset but has only been well characterized in this species, probably due to its small size and rapid evolution [42,43]. However, orthologues for its receptor, the sex peptide receptor (*SPR*), are very well conserved across insects and have been found in a number of species from other taxa, including Lepidoptera such as *Helicoverpa armigera* (Lepidoptera: Noctuidae) and *Spodoptera litura* [44,45]. *SPR* mediates the post-mating switch in the reproductive behavior of mated females, such as reduced receptivity and increased oviposition [43,45].

In this study, the remating propensity of female *S. litura* (with normal males) that were initially mated with irradiated sub-sterile males or their F_1_ male progeny was assessed. The effect of post-mating delay after first mating on the remating behavior of the mated females with unirradiated males was also evaluated. The study was extended to investigate the ejaculate quality of the irradiated males and their F_1_ male progeny with respect to their initial mating, and the influence was correlated with the remating propensity of the mated females with normal males and associated reproductive viability.

## 2. Materials and Methods

### 2.1. Insect Rearing

Adult *S. litura* moths were collected in agricultural fields of the Indian Agricultural Research Institute, New Delhi, and the colony was maintained in the laboratory of the Department of Zoology, University of Delhi, at a temperature of 27 ± 1 °C, at 75 ± 5% relative humidity and under a photoperiod of 12:12 h L:D. The larvae were reared on a semi-synthetic diet [46].

### 2.2. Insect Irradiation

The male moths were irradiated using a Co^60^ Gamma Chamber-5000 (Gamma-5000 irradiator, Board of Radiation and Isotope Technology, Mumbai, India) that was placed in the radiobiological unit of the Institute of Nuclear Medicine and Allied Sciences (INMAS), Ministry of Defense, Delhi-110054. The dose rate of the gamma cells ranged between 10.4 Gy/min and 5.9 Gy/min. Freshly emerged male adults (0–1 d old) were exposed to a sub-sterilizing dose of 130 Gy, proposed for the F_1_ sterility technique against *S. litura* by Seth and Sehgal (1993) [47] and Seth and Sharma (2001) [46]. The irradiated male moths (0–1 d old) were paired with unirradiated females (0–1 d old) to obtain F_1_ male progeny.

### 2.3. Precopulatory, Mating and Reproductive Behavior of Mated Female Moths

Virgin female moths, aged 0–1 d old, were paired with males (0–1 d old), obtaining 12–15 pairs per cage made of Perspex and nylon (45 × 30 × 30 cm), in the following combinations: (i) unirradiated females (normal) (N♀) × unirradiated males (normal) (N♂) as untreated control; (ii) N♀ × irradiated males (T♂_130_) (treated with 130 Gy as 0–1-day-old adults); (iii) N♀ × F_1_ male moths (progeny of N♀ × T♂ cross).

During the scotophase, the cages were inspected every 15 min under red light, to observe female ‘calling’ behavior and the formation of mating pairs. The calling behavior of a virgin female was recorded by examining the posture of the female’s abdomen during calling. The female showed slightly raised wings and an exposed pheromone gland in the first two hours of the scotophase. Mating pairs found were transferred into 500 mL glass jars with porous lids and observed until the mating pairs were dislodged or separated. Records were kept of the copulation duration. After mating, the male was removed and the female was left in the cage with 10% honey as food. An average reading of the duration of each female’s calling from each cage (comprising 3–4 pairs) constituted one replicate.

The mating success of virgin females paired with irradiated males and F_1_ males was recorded and compared with the control. To assess mating success, each replicate comprised 12–15 pairs of moths in a specific regimen.

The mated females, placed in a Perspex–nylon cage, were provided with castor leaves, *Ricinus communis*, as an ovipositional trap and 10% honey as an adult food source. The egg masses were collected from each cage during the oviposition period. For each experimental regimen, an average reading from 10 egg samples from each cage (comprising 3–4 pairs) for testing hatchability constituted one replicate. The experiments were repeated 10 times for each condition.

### 2.4. Insemination Quality of Irradiated Moths and F_1_ Progeny

The insemination quality of 130 Gy-irradiated males and F_1_ males when paired with an unirradiated female was evaluated in terms of spermatophore mass, its total protein content and the number of sperm.

For this study, the unirradiated females (N♀) were paired with irradiated male moths (N♀ × T♂_130_), F_1_ males (derived from a cross of irradiated males and unirradiated females) (N♀ × 130 F_1_♂) and unirradiated male moths (N♀ × N♂) in different regimens. Upon termination of copulation, the females were removed and kept in a freezer before extraction of the spermatophore from the female’s bursa copulatrix. The fresh weights of the spermatophores were weighed and each one was put into an individual plastic 0.2 mL Tarson Eppendorf tube. The tubes containing the ejaculates (extracted from the spermatophores) were transferred to a freezer (LG 437) where they were stored at −20 °C. The ejaculates were assayed for their protein concentration using the BCA Protein Assay Kit (Thermo Fisher Scientific, Wilmington, DE, USA, Cat. No. 23225) [48] with bovine serum albumin (BSA) as a standard.

To determine the total number of eupyrene sperm bundles, the content of the spermatophore was spread on a microscope slide for examination using a phase-contrast microscope (Nikon TMS, Melville, NY, USA). To count the apyrene sperm (loose sperm), the content of the spermatophore was diluted by adding a known volume of Belar’s saline (6 g NaCl, 0.2 g KCl, 0.2 g CaCl_2_, 0.2 g Na_2_CO_3_ and water to make 1 L) in Eppendorf tubes, and 10 μL subsamples were removed from the diluted spermatophore content to count the apyrene sperm number using a Neubauer hemocytometer (Fein-Optix Blankenburg, GDR); an average reading of these ten samples constituted one replicate. Apyrene sperm could be distinguished from eupyrene sperm, being considerably thinner, shorter in length and more undulating in shape. Ten replicates for each regimen were studied. To neutralize the impact of male body size on the weight of the spermatophore and its protein content, emerged male moths with a mean weight of 97.34 ± 0.73 mg were used for experimentation.

### 2.5. Expression of Sex Peptide Receptor (SPR) in Mated Females

The female moths (0–1 d) were allowed to mate with unirradiated or irradiated males (0–1 d), and six hours after the initial mating, the head regions of mated and virgin females were dissected and sampled. For each combination, six replicates were carried out. The tissues were stored in Trizol reagent (Thermo Fisher Scientific, Wilmington, DE, USA, Cat. No. 15596018) at −80 °C (New Brunswick Scientific type U570, Eppendorf AG, Hamburg, Germany) until further processing.

Total RNA was isolated from the sampled head tissues using the phenol–chloroform extraction method [49], and the concentration of RNA was determined using a NanoDrop ^TM^ 2000C (Thermo Fischer Scientific, Wilmington, DE, USA). RNA samples with A260/280 ratios of 1.8–2.0 was considered pure RNA. The RNA was pre-treated with DNase (M6101; Promega, Madison, WI, USA) to remove any DNA contamination, and then, it was reverse transcribed using a Revert Aid first strand cDNA synthesis Kit (K1622; Thermo Fisher scientific, Wilmington, DE, USA). Afterwards the cDNA was amplified with the reference gene β actin, and the quality of cDNA was checked on 0.1% agarose gel.

To assess the mRNA expression of the *SPR* gene, gene-specific primers were designed from the cDNA sequences using the Eurofins MWG Operon primer design (http://www.operon.com/tools/oligo-analysis-tool.aspx, accessed on 21 November 2019) online program. The primers used were as follows—for the *SPR*: sense primer 5′-GGCAGTTTAGGGAGACGTTTA-3′ and antisense primer 5′-CTACTAGAGCCGCCATTCTTAC-3′ (Accession no. XM_022973010.1), and for the elongation factor 1 alpha (*EF1α*): sense primer 5′-GACAAACGTACCCATCGAGAAG-3′ and antisense primer 5′-GATACCAGCCTCGAACTCAC-3′ (Accession no. XM_022965580.1).

The mRNA expressions were evaluated via qualitative real-time PCR (qRT-PCR) run on a thermocycler (ViiA7, Applied Biosystems, Foster City, CA) using SYBR green. Both the sample *(SPR)* and reference gene (*EF1α*) were run in duplicate, and the relative mRNA expression level was calculated as the fold change, the 2^−(ΔΔCT)^ value [50]. Firstly, the Δ*C*_T_ value was calculated by subtracting the threshold cycle (*C*_T_) of the reference gene from the target gene (*C*_T_ (target)–*C*_T_ (reference)), and then, Δ*C*_T_ was normalized against the ΔC_T_ value of the pool sample that consisted of a mix of cDNA of all samples, which was composed of a mixture of cDNA of female samples of all regimens involving matings with irradiated males and unirradiated males; this gave the ΔΔ*C*_T_ value, and the negative value of this powered to 2 (2^−ΔΔCT^) was plotted.

### 2.6. Remating Behavior of Female Moths

The proportion of female calling, the duration of the calling and the remating propensity of mated females with unirradiated males were assessed to determine whether the status of the male during first mating had any influence on the female’s remating behavior. 0–1 d old virgin female moths (*n* = 12–15) were put in a cage made of Perspex and nylon (45 × 30 × 30 cm) and were allowed to mate with 0–1 d old males in the following combinations: (i) unirradiated females (N♀) × unirradiated males (N♂); (ii) N♀ × irradiated (130 Gy) males (T♂_130_); (iii) N♀ × F_1_ male moths (derived from the mating of a virgin untreated female parent with a 130 Gy-irradiated male parent). All mated females were offered a second mating opportunity with unirradiated males (N♂) (0–1 d old) at different intervals of 24 h, 48 h and 72 h post-first mating. The females were observed for 3 consecutive days.

On each day, the females that had re-mated were recorded and removed from the cage in each regimen, and the remaining females were evaluated for remating vis-a-vis different intervals post-first mating. All remated females were dissected within 15 min of uncoupling to obtain spermatophores, which were weighed as mentioned above. For remating trials, 12–15 females already mated to a given male type (N♂, T♂_130_, or 130 F_1_♂s) were placed with an equal number of unirradiated males in a cage (45 × 30 × 30 cm), 2 to 3 h before scotophase, and the number of females involved in remating were observed per cage during scotophase (18:00–6:00).

As for the original mating experiments, the number of calling females and their calling duration were recorded during remating of the females with unirradiated males. The calling behavior of mated females was documented in different regimens of the post-first mating interval of 24, 48 and 72 h during the first two hours of scotophase, which was the peak calling phase, as also reported by Lu [51]. The replication was performed ten times, and an average reading of such behaviors in a cohort of 3–4 females paired with normal or irradiated males (T♂_130_ or 130 F_1_♂) constituted one replicate. The copulation duration of already mated normal females during their remating with unirradiated males was also recorded. An average reading for copulation duration from each experimental cage (comprising 3–4 pairs) constituted one replicate, and ten replicates were carried out for each regimen. An average reading for the fertility of 5–7 egg batches from an experimental cage of each remating regimen of mated females with virgin unirradiated males constituted one replicate, and the experiment was repeated 10 times for each regimen. Oviposition trials were conducted as detailed in Section 2.3.

### 2.7. Statistical Analysis

Data on females’ premating and mating behavior and egg fertility were analyzed after first mating with irradiated males or their F_1_ progeny, using one-way analysis of variance (ANOVA) followed by Tukey’s test. One-way ANOVA was also performed to test the effects of male irradiation on spermatophore mass, protein content, sperm count and gene expression, followed by Tukey’s test. Student’s *t*-test was used to examine the effect of mating status (virgin female vs. mated female) on *SPR* gene expression. Data on calling behavior, and other reproductive features during females’ remating with unirradiated males in relation to consecutive mating intervals, were analyzed using two-way ANOVA followed by Tukey’s test. The statistical analyses were carried out using the GraphPad prism software program, version 9.3.1 (San Diego, CA, USA). For statistical significance, the *p*-value was set at 0.05. Unless stated otherwise, all values reported are means ± S.E. The percentage data were arcsine transformed before the ANOVA and the data in the graph are back transformations.

## 3. Results

### 3.1. Female Premating and Mating Behavior after Pairing with Irradiated Males and F_1_ Male Progeny

Various parameters were assessed by pairing 130 Gy-treated parent males or their F_1_ offspring (derived from irradiated male parents crossed with unirradiated females) with unirradiated females. The proportion of calling females (F_(2,27)_ = 0.18; *p* = 0.84) (Figure 1a) and calling duration (F_(2,27)_ = 0.38; *p* = 0.69) (Figure 1b) were similar for females paired with irradiated male parents or their F_1_ male progeny, and unirradiated males.

Copulation duration was significantly influenced by male irradiation status (F_(2,27)_ = 45.02, *p* = 0.0002) (Figure 1c). Irradiated males and their F_1_ male progeny mated significantly longer than their unirradiated counterparts, i.e., copulation time was extended by 40% (69.1 min) and by 57% (77.7 min) for irradiated males and their F_1_ male progeny, respectively.

Mating success was significantly affected by the irradiation status of the males (F_(2,27)_ = 45.02, *p* = 0.0002) (Figure 1d). F_1_ males were the least successful in mating, followed by irradiated parent males and unirradiated males. The mating success of irradiated parent males and F_1_ males was 74% and 69%, respectively, compared with 82% for the unirradiated control matings.

### 3.2. Insemination Quality of Irradiated Moths and F_1_ Progeny and Its Correlation with Fertility

The weight of spermatophores, their total protein content and the proportion of sperm transferred during the mating of irradiated male moths or their F_1_ male progeny were assessed to determine the insemination quality of these treated males. The average weight of the spermatophores of unirradiated males that were transferred to females after successful mating was 3.26 (±0.18) mg. The irradiation status of males significantly affected the spermatophore weight (F_(2,27)_ = 8.91; *p* < 0.001) (Figure 2a) as well as its total protein content (F_(2,27)_ = 7.74; *p* = 0.02) (Figure 2b). The average weight of the spermatophores transferred by F_1_ males was 16% and 7% lighter in comparison with those of unirradiated males (3.26 mg) and irradiated parent males (2.96 mg), respectively. Similarly, the protein content of the spermatophores of F_1_ males was 31.5% and 12.6% lower in comparison with those of unirradiated males (0.95 µg/µL) and irradiated parent males (0.83 µg/µL), respectively. A small but nonsignificant decrease was observed in the weight and protein content of the spermatophores of irradiated parent males and unirradiated males.

The irradiation status of the males had a significant effect on number of eupyrene sperm bundles (F_(2,27)_ = 86.5; *p* < 0.001) (Figure 2c) as well as amount of apyrene sperm (F_(2,27)_ = 49.97; *p* < 0.001) (Figure 2d) transferred to bursa copulatrix of females. The number of eupyrene sperm bundles and the amount of apyrene sperm transferred to females by irradiated parent males (308.3 eupyrene sperm bundles and 229.5 *×* 10^3^ apyrene sperm) or their F_1_ male progeny (198.9 eupyrene sperm bundles and 188.8 *×* 10^3^ apyrene sperm) were significantly lower compared with those from unirradiated males (347.8 eupyrene sperm bundles and 291.3 *×* 10^3^ apyrene sperm). These results suggested that irradiation reduced the males’ ejaculate quality and the effect was more prominent in F_1_ males (derived from 130 Gy-irradiated male parents) than in the irradiated parents.

Male radiation treatments (F_(2,27)_ = 144.3; *p* < 0.001) (Figure 2e) also caused a significant reduction in the egg hatch percentage of females that had mated with treated males, with the greatest reduction occurring when paired with F_1_ males. The egg hatch percentage was reduced by 48.2% and 78.3% in the mating pairs involving irradiated sub-sterile male parents (45.2%) and F_1_ males (18.9%), respectively, compared with that of unirradiated insects (87.4%) (N♂ × N♀).

Further, the expression pattern of the sex-peptide receptor (*SPR*) gene was also studied in female moths (adult head region) after mating with unirradiated males, irradiated males and F_1_ males. The matings resulted in significant upregulation of *SPR* expression in females (t = 8.969, df = 10, *p* < 0.0001) (Figure 3a). Male irradiation status (*F*_(2,15)_ = 10.47; *p* = 0.0014) (Figure 3b) had a significant effect on *SPR* expression in the mated females. The relative gene expression of an unirradiated female crossed with a 130 Gy-irradiated parent male was not significantly different from the unirradiated control. However, in the corresponding cross with F_1_ males, the relative gene expression was significantly decreased in females in comparison to females in other both regimens when mating occurred with unirradiated males or 130 Gy-irradiated parent males.

### 3.3. Remating Behavior of Mated Females in Response to First Mating with Irradiated Males or Their F_1_ Progeny

The effects of first mate (male) condition (irradiation status) and post-mating interval after first mating were studied on female calling behavior (proportion of female calling and calling duration), remating propensity, copulation duration, the weight of the spermatophores transferred during remating with unirradiated males and eventual egg fertility.

In the unirradiated control group (N♀ × N♂ × N♂), the sequential matings of unirradiated females with unirradiated males showed that 18% of the mated females exhibited calling behavior for remating at 24 h after initial mating. This proportion of calling females was significantly increased when the interval between the two consecutive matings was increased to 48 h (~31%) and 72 h (~34%). In a mating cross of N♀ with a T♂_130_ followed by a second mating with an unirradiated male (N♂), and in a mating cross of a N♀ with a 130 F_1_♂ followed by a second mating with N♂, the proportion of females calling was also significantly increased with an increase in the interval between two consecutive matings. The increase in the proportion of females calling was influenced by the male irradiation status involved in first mating, with the impact being greater in the regimen of N♀ × 130 F_1_♂ × N♂, where F_1_ males were crossed with unirradiated females (Figure 4a).

The irradiation status of the male in the first mating (F_(2,81)_ = 13.91; *p* < 0.001), as well as the interval between two matings (F_(2,81)_ = 26.32; *p* < 0.001), had a significant impact on the proportion of females calling, although the interaction between these variables was not significant (F_(4,81)_ = 1.74; *p* = 0.16). Irrespective of the irradiation history of the first male mate, the proportion of females calling during remating was increased with an increase in post-mating interval. 

In the control group, the unirradiated females called for about 22 min during the second mating with unirradiated males, and the calling duration was increased with an increasing interval between the consecutive matings. The calling duration during the second mating was 24.2 min and 25.8 min with an inter-mating interval of 48 h and 72 h, respectively. In the first mating with sub-sterilized males or their F_1_ male progeny, the calling duration of females during the second mating was significantly affected by first mate irradiation status (F_(2,81)_ = 29.42, *p* < 0.001) and the interval between the two matings (F_(2,81)_ = 24.19, *p* < 0.001). The interactions between these variables (irradiation status of first male mate and inter-mating interval) were not significant (F_(4,81)_ = 0.46, *p* = 0.76) (Figure 4b).

Females initially mated with F_1_ males called for significantly longer during the second mating than the unirradiated controls (females that were first mated with an unirradiated male). Further, calling duration during second mating increased proportionally with increasing in inter-mating intervals (24 h, 48 h or 72 h) in both the regimens, involving first mating with an irradiated male or an F_1_ male (N♀ × T♂_130_ × N♂ and N♀ × F_1_♂ × N♂); the impact was greater in the latter case.

In the control group, the remating propensity of unirradiated female was 12% with an unirradiated male, and it was increased up to 22–24% with an increase in the interval between the two matings. A similar trend was observed when the first male mate was either sub-sterilized or an F_1_ male. Females initially mated with an F_1_ male were twice as receptive to new mates (unirradiated male moths) compared with the control group (N♀ × N♂ × N♂) at the inter-mating interval of 24 h. Remating propensity was more influenced (increased) in females first mated with an irradiated male or F_1_ male progeny compared to those crossed with an unirradiated male. Male irradiation status in the first cross (F_(2,81)_ = 6.83, *p* = 0.003) and time interval since first mating (F_(2,81)_ = 9.21, *p* = 0.0006) had a significant effect on female remating tendency (Figure 4c). The interaction between these variables (irradiation status of first male mate and inter-mating interval) was also significant (F_(4,81)_ = 2.64, *p* = 0.04). Overall, the remating propensity of females was found to be increased in response to the first mating of unirradiated females with F_1_ males than irradiated parent males or controls (unirradiated males).

In the control group, the mating duration of an unirradiated mated female during her second mating with an unirradiated male was around 67 min. The mating period of female moths was not affected by the irradiation status of first male mate (F_(2,81) =_ 1.17; *p* = 0.35) and inter-mating interval (F_(2,81) =_ 0.37; *p* = 0.69) among all the experimental regimens (Figure 4d).

The weight of a spermatophore transferred to an unirradiated female during second mating with an unirradiated male was 3.13 mg. Spermatophore weight was not influenced by the irradiation status of the first male mate (F_(2,81) =_ 0.76; *p* = 0.48), but a slight impact was noticed when the interval between two matings was increased. (F_(2,81)_ = 11.59, *p* = 0.006) (Figure 4e).

When an untreated female that had previously mated with an irradiated or an F_1_ male was remated with an unirradiated male, the egg fertility was increased with an increasing interval between the two matings, unlike in the control group (F_(2,81) =_ 39.13; *p* < 0.001) (Figure 4f). At the same time, it was interesting to record that egg fertility due to remating was significantly lower in cases of first male mates being F_1_ male moths, followed by irradiated male parents and controls (unirradiated males) (F_(2,81) =_ 24.79; *p* = 0.0002).

## 4. Discussion

In the SIT/IS for Lepidopteran pests, multiple matings of wild females are an important factor that could influence the reproductive success of released irradiated male moths. Although female monogamy is not crucial for the SIT, its success might be affected by various associated factors: 1. multiple mating of females with sterile and/or wild males at a different rates; 2. sperm selection after multiple matings [52,53]; and 3. chemicals transferred by the male during mating. With respect to the last factor, irradiated males that fail to transfer the required chemicals might increase the incidence of remating of wild females [54,55]. Multiple mating in female *S. litura* is common, and female mating success, remating frequency and fertility could be significantly affected in sequential matings that involve irradiated and unirradiated (wild) males [16]. Female remating, together with dispersal, mating competitiveness, sperm transfer and sperm competition, may affect the induction of sterility in wild populations [56].

Therefore, this study was carried out to determine the receptivity of females, along with mating behavior and reproductive success, during remating with unirradiated males preceded by initial mating with irradiated males or their F_1_ male progeny.

### 4.1. Premating and Mating Behavior after Pairing with Irradiated Males and F_1_ Male Progeny

During first mating, it was observed that female calling was unaffected but mating duration and mating success were affected by the irradiation status of the male mate. Mating success in the present study was adversely affected in F_1_ males compared with irradiated male parents when crossed with unirradiated females, and this is in consonance with similar results reported by Seth and Sharma [35]. The reduced mating success could be correlated with the flight ability of F_1_ male progeny derived from 130 Gy-irradiated male moths of *S. litura* [16]. Additionally, the increased mating latency of irradiated males might have also contributed to the reduced mating success [22]. The acceptance rate of mass-reared radio-sterilized males by wild normal (unirradiated) females could also add to the rate of mating success. Lance et al. (2000) reported that in *Ceratitis capitata* (Diptera: Tephritidae), females were more likely to copulate with fertile males than with sterile males [57].

Prolonged copulation duration in moths was correlated with radiation dose [25,58,59] and male mating status [25]. In the current study, the mating duration was longer when irradiated or F_1_ males were mated to unirradiated females, which could suggest insufficient quality of the ejaculate or difficultly in transferring a spermatophore during mating [60]. Alternatively, involving the female in extended copulation might effectively limit the female to copulating only once a day [61]. Long mating time might be a strategy used by males to prevent the remating of females until after the transferred sperm have migrated out of the spermatophore [11]. Such delayed mating would reduce the probability of last male sperm precedence due to the displacement of previous sperm. Changes in the mating behavior of irradiated or F_1_ males (with normal females) might be attributed to the quality of the seminal fluid.

### 4.2. Insemination Quality of Irradiated Males and F_1_ Male Progeny and Its Correlation with Fertility

Mated females undergo a range of physiological and behavioral changes, including reduced receptivity to remating, the stimulation of oviposition, changes in female flight and feeding behavior and the modulation of sperm storage. Many of these post-mating changes in the female are modulated by the ejaculate transferred by the male moth, which contains sperm, seminal fluid proteins and other components produced in the male reproductive tract [26,28,41]. After mating, female receptivity to mating with a new male is inhibited for a variable length of time. The inhibition to remating usually results from neural stimulation of the CNS by sperm and/or accessory gland secretions or the presence of the spermatophore in the bursa [62,63].

To test the hypothesis that insemination quality might impact upon mating behavior and reproductive success, the influence of irradiation was assessed on the quality of male insemination by examining the weight of the spermatophores, their protein content and sperm quantity. In the present study, it was observed that F_1_ males transferred relatively poor-quality ejaculate, as the spermatophores were lighter with less protein and fewer sperm than the ones transferred by unirradiated males, whereas a slight but insignificant decrease in ejaculate quality in F_1_ males was observed when compared to the ejaculate quality from irradiated sub-sterile parents. The smaller spermatophore mass in F_1_ male moths might be due to the reduced growth of the internal organs as a result of delayed growth and development in the F_1_ progeny of *S. litura*, as reported by Seth et al. (2000) [64]. The spermatophore size could be correlated with its insemination quantity and quality, which might affect the mating dynamics. For instance, larger spermatophores have been associated with the induction of a longer refractory period in the female [38,65]. This delay in remating could thus be associated with degradation of the spermatophore [66,67] and/or the weight of the spermatophore, which would induce mechanical stimulation by acting on stretch receptors in the sperm storage organ [68].

The F_1_ males, derived from the mating of an irradiated sub-sterilized male with an unirradiated female, transferred significantly fewer sperm, which was also reported in *Lymantria dispar* [69]. The quantity of sperm transferred by males could be correlated with radiation dose as well as male mating status [25]. For instance, increasing gamma dose affected the number of both eupyrene and apyrene sperm in *Ephestia kuehniella* (Lepidoptera: Pyralidae) [59] and *S. litura* [16,25]. The smaller amount of sperm transferred might be attributed to the effect of irradiation on sperm production and/or sperm descent [22]. A radio-sterilization-associated decrease in the sperm quantity transferred to females could reduce sterile male competitiveness by increasing the incidence of remating tendency, which would reduce the fertility of females. The fertility of females after the first mating with F_1_ male progeny was evidently lower than the fertility observed after the first mating with irradiated male parents in the present study. Taylor (1967) showed that the efficiency of fecundity in *Atteva punctella* (Lepidoptera: Yponomeutidae) was inversely correlated with the receptivity of mated females and, as a result, with the frequency of multiple mating, too [70]. 

The expression patterns of the *S. litura* sex peptide receptor (*SPR)* gene were also studied in the head region of the adult moths in relation to their mating status. The relative expression levels of *SPR* showed a significant increase after mating similar to that reported by Li et al. [45] in *S. litura*. Furthermore, the expression patterns of the *SPR* gene after mating with irradiated males and F_1_ male progeny were studied. The relative *SPR* expression level was decreased significantly in females mated to F_1_ males. This might be correlated with the smaller weight of spermatophores with low protein content transferred by F_1_ males and the ensuing high receptivity in the already mated females. Li et al. showed, using *SplSPR* dsRNA, that *SplSPR* could function in detecting factors from MAG secretions towards stimulating oviposition and egg maturation in *S. litura* [45].

### 4.3. Remating Behavior in Response to First Mating with Irradiated Males or Their F_1_ Progeny

For the SIT/IS, a differential rate of remating in normal females first mated with an irradiated male might compromise the outcome of such a program [71,72]. It might be possible that for a female moth who, after initially mating with an irradiated male, becomes sterile, a second mating with a fertile male could potentially restore the fertility of the female; therefore, multiple mating of females of the target pest species could affect the efficacy of control methods such as the SIT/IS.

Receptive females attract males by releasing pheromones during a characteristic ‘calling’ phase. Mating resulting in a temporary or long-term loss of sexual receptivity or reduced sexual attraction in many insects could be partly due to interruptions in pheromone production (pheromonostasis) and/or the absence of calling activity [26,27,73,74]. Mating has been reported to trigger a remarkable alteration in the reproductive behavior of both males and females in many insects, such as *S. litura* [45], *E. kuehniella* [75] and *H. armigera* [44] from Lepidoptera and *D. melanogaster* [42] from Diptera. Mated females exhibited less calling behavior and were resistant to remating attempts [28,73,76].

In the present context, during remating, the proportion and calling duration of the calling females initially mated to F_1_ males were significantly higher those of females that had undergone a first mating with an unirradiated male. Also, as the intervals between the two matings were increased, there was an increase in the proportion of calling females irrespective of the irradiation status of the first male. According to a study on *H. zea*, MAG was found to produce a pheromostatic peptide (PSP) that temporarily depleted females’ pheromones, making them unreceptive to new mates [77]. Lu et al. (2017) showed that mating resulted in a higher sex pheromone titer in the pheromone gland (PG) of *S. litura* [51]. Later, Xu et al. (2019) attributed the suppression of female calling behavior to mating and MAG factors, which resulted in higher sex pheromone titers in mated *S. litura* females, presumably due to a lower release of pheromones [78]. This implied that irradiation probably influenced MAG factors that might have consequently failed to suppress female calling behavior effectively when mated to F_1_ males due to insufficient or less effective ejaculate.

While investigating female remating propensity in *S. litura* in the present study, it was observed that the ability of F_1_ males crossed with unirradiated females to inhibit female remating propensity 24 h post first mating was drastically lower than the ability of unirradiated males involved in first mating, although these differences were diminished at 48 h and 72 h post first mating. Likewise, the *C. capitata* Vienna 4 strain irradiated with 150 Gy was less able to inhibit female remating 24 h after first copulation, while 48 h and 72 h after first mating, the proportion of female remating was not influenced by first male mate status [79]. Similarly, female *L. dispar* mated to F_1_ sterile moths remated significantly more than those mated to unirradiated moths [80]. Higher remating was also observed in *Phthorimaea operculella* (Lepidoptera: Gelechiidae) [81], *Anastrepha serpentine* (Diptera: Tephritidae) [82] and *Zeugodacus cucurbitae* (Diptera: Tephritidae) [83] females first mated to sterile males, whereas remating levels did not differ in *A. fraterculus* (Diptera: Tephritidae) [84], *A. ludens* (Diptera: Tephritidae) [85] and *D. suzukii* (Diptera: Drosophilidae) [86] females first mated to wild or sterile males. In the present study, the remating propensity of females crossed with irradiated sub-sterilized male parents followed a similar trend to those crossed with unirradiated male moths, as also observed in case of *H. armigeria* [87] and *S.litura* [16], whereas the females mated to F_1_ male progeny remated earlier and more frequently. The increase in the remating tendency of untreated females with F_1_ males appeared to be correlated with their increased calling behavior, decreased mating success and decreased oviposition. Increased female calling behavior with increasing intervals between the two matings resulted in greater female receptivity. The higher receptivity of females during remating with increasing inter-mating intervals could also be due to a decline in sperm supply and a diminished inhibitory effect of the accessory gland (MAG) secretions of the males during first mating [88,89]. Sexual receptivity during remating with irradiated/F_1_ male moths appeared to be negatively correlated with the spermatophore size experienced in their first mating in the present study. This indicates that there might be a threshold for the weight of the spermatophore, below which females would become receptive sooner. Sugawara (1979) suggested that receptiveness might get triggered mechanically by the activation of stretch receptors in the sperm storage organ [68].

The increase in the sexual receptivity of the females 24 h after the initial mating with an F_1_ male might be related to a lower weight of the spermatophores and low ejaculate volume. This result suggested that the refractory period of the females, in its initial phase, seemed to be induced mechanically by the spermatophore acting on stretch receptors in the sperm storage organs. Later, MAG secretions and sperm might influence the refractory period in the long term by inducing a change in female behavior [26,27]. These findings support the idea that ejaculate quality influences female remating tendency. A similar finding was also reported by Yu et al. [28] in *S. litura* and Brent [67] in *Lygus hesperus* (Hemiptera: Miridae). In *S. litura*, the body weights of the female and second male mate were also linked to female remating tendency; the heavier the female and second male, the more likely it was that the female would remate [15].

The irradiation status of the first male mate and the interval between the two matings did not influence mating duration and the weight of the spermatophore during the second mating of unirradiated females with unirradiated males. This indicates that unirradiated males transferred similar ejaculate irrespective of the irradiation status of the first male mate or the interval between the two matings. However, males transferred smaller spermatophores to aged females [77]. Since males do not produce unlimited secretions, they might benefit from discriminating among females and selecting mates that would maximize egg production. Egg production was found to decrease as females aged in several Lepidoptera [90,91]; hence, males might copulate preferentially with younger females.

Along with female receptivity, it is important to understand the pattern of sperm utilization in *S. litura* because of incidences of multiple mating. The interval between matings is a temporal factor that influences sperm precedence [92,93,94]. According to the ‘passive sperm loss’ hypothesis, as the interval between matings increases, passive loss of the first male’s sperm from the female reproductive system causes disproportionate use of the sperm of the second male mate [95,96]. In the present investigations, in female moths that had previously mated with irradiated or F_1_ males, the egg hatch percentage after a second mating with unirradiated males was increased with an increase in the interval between the two matings, which suggested a different pattern of sperm usage after remating. The sperm usage patterns seen in this study with varying intervals in between the two matings are in consonance with other similar studies [97,98]. The last male sperm precedence, related to sperm usage pattern, might place the male under selection pressure to evolve characteristics that would delay or prevent females from remating [99]. The fertility of the female seems to depend on the first mate’s and second mate’s status as well as the interval between first and second mating. Female polyandry enables females to rescue their fertility when male reproductive function is compromised by gamma radiation.

## 5. Conclusions

It is evident from this study that the intervals between the two matings and the irradiation status of the first male mate were presumably crucial factors influencing the tendency of females to remate. The females that had been first mated to irradiated males remated earlier and more frequently, and when the gap was increased between the first and subsequent matings, the propensity of the female to remate was increased. The radiation-induced changes in the ejaculate quality of partially sterile males were not sufficient to influence female receptivity. However, the slight but significant decrease in the ejaculate quality of the F_1_ males, along with the smaller spermatophore transferred, might be the vital factors resulting in increased receptivity in the mated females and their induced reproductive sterility.

The irradiation status of male moths had a significant bearing on female receptivity and its remating propensity in *Spodoptera litura*, which were found to be influenced by the insemination quality of irradiated male parents and their F_1_ male progeny in conjunction with post-first mating delay. These parameters are crucial considerations in modeling the logistic operation of the IS technique for Lepidopteran pest management.

## Figures and Tables

**Figure 1 insects-14-00651-f001:**
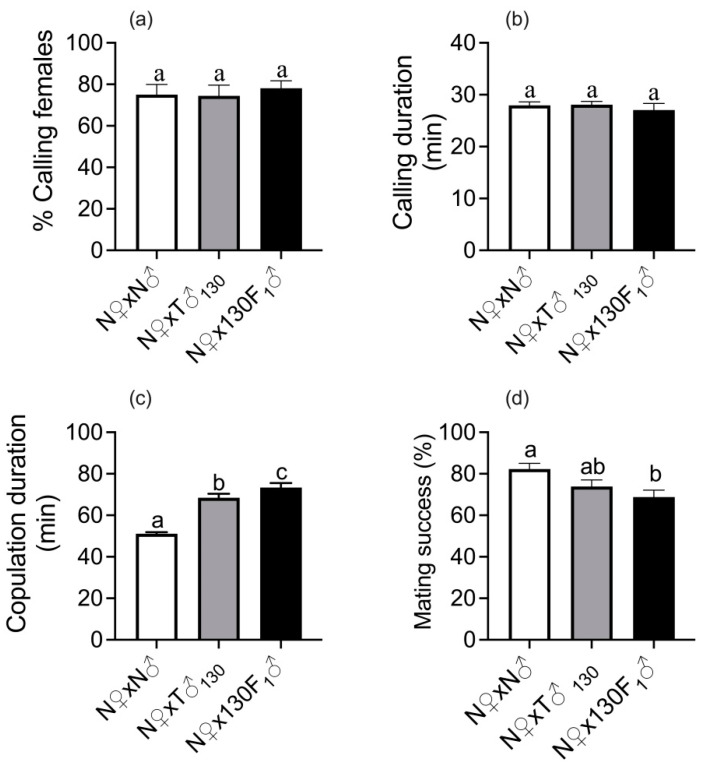
Effect of irradiation status of first male mate on premating and mating behavior. (**a**) Calling females (%), (**b**) calling duration (min), (**c**) copulation duration (min) and (**d**) mating success (%). Means ± S.E followed by same small letters are not significantly different at *p* < 0.05 level (one-way ANOVA followed by Tukey’s post hoc test); *n* = 10, where an average reading in a group of 3–4 pairs per cage for calling behavior and copulation duration constituted one replicate, whereas 12–15 pairs per cage constituted one replicate for assessing mating success. Percent data were arcsine transformed before ANOVA, but data in graph are back transformations. N♀—unirradiated female, N♂—unirradiated male, T♂_130_—male irradiated with sub-sterilizing dose of 130 Gy, and 130 F_1_♂— F_1_ male progeny derived from sub-sterile male.

**Figure 2 insects-14-00651-f002:**
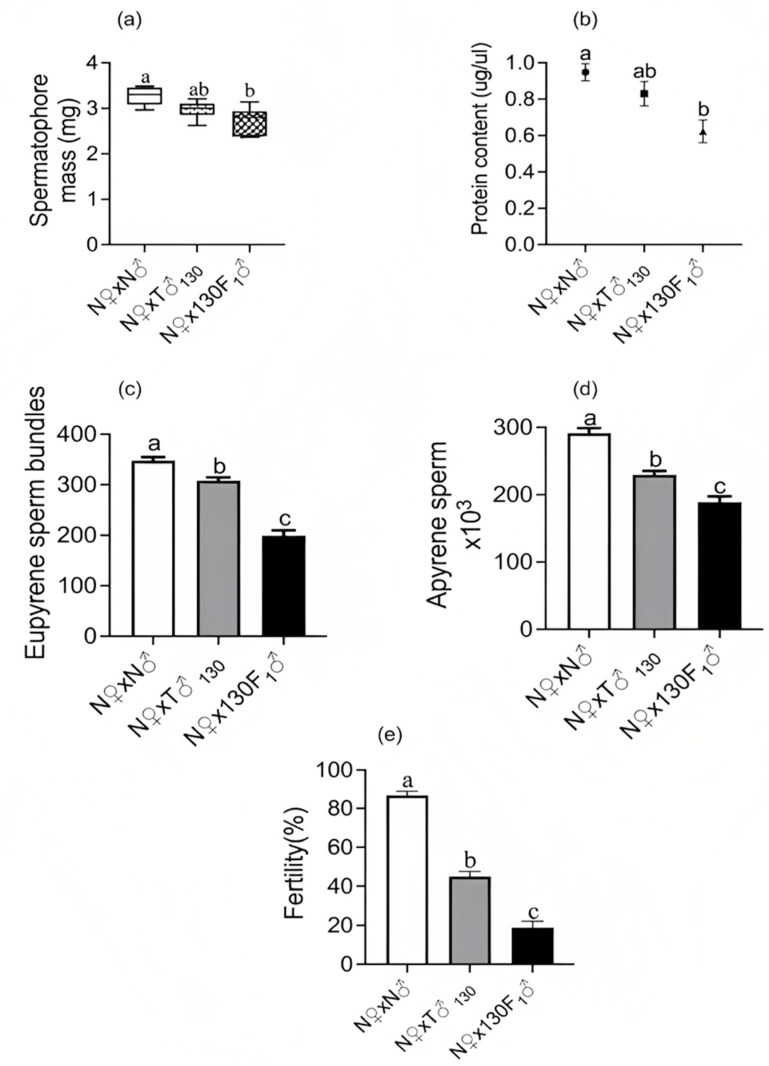
Effect of first male mate irradiation status on its insemination quality in relation to egg fertility in mated females. (**a**) Spermatophore mass (mg), (**b**) spermatophore protein content (μg/μL), (**c**) eupyrene sperm bundles, (**d**) apyrene sperm and (**e**) fertility (%). Means ± S.E followed by same small letters are not significantly different at *p* < 0.05 level (one-way ANOVA followed by Tukey’s post hoc test); *n* = 10, where an average reading in a group of 4–5 pairs per cage constituted one replicate. Percent data were arcsine transformed before ANOVA, but data in graph are back transformations. N♀—unirradiated female, N♂—unirradiated male, T♂_130_—male irradiated with sub-sterilizing dose of 130 Gy, and 130 F_1_♂— F_1_ male progeny derived from sub-sterile male.

**Figure 3 insects-14-00651-f003:**
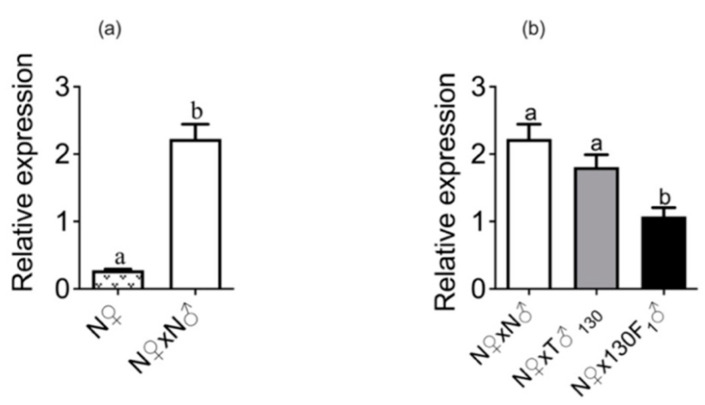
Effect of irradiated males and their F_1_ male progeny on *SPR* expression in mated females. (**a**) Relative gene expression in females before and after mating with normal (unirradiated) males, (**b**) relative gene expression in females after mating with treated males (irradiated males or their F_1_ male progeny). Means ± S.E followed by same small letters are not significantly different at *p* < 0.05 level; (**a**) *t*-test, (**b**) one-way ANOVA followed by Tukey’s post hoc test. N♀—unirradiated female, N♂—unirradiated male, T♂_130_—male irradiated with sub-sterilizing dose of 130 Gy, and 130 F_1_♂—F_1_ male progeny derived from sub-sterile male.

**Figure 4 insects-14-00651-f004:**
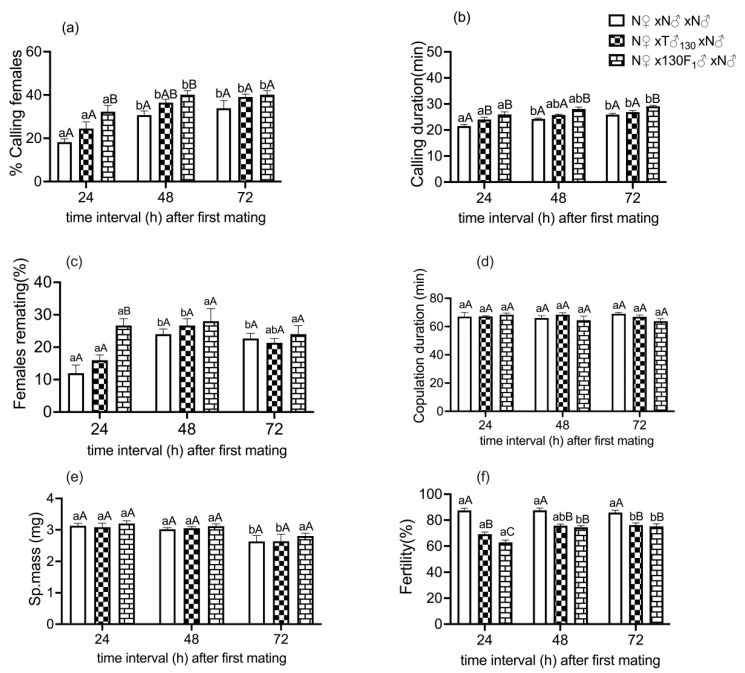
Effect of first mate irradiation status and post-mating interval on female remating behavior. (**a**) % calling females, (**b**) calling duration (min), (**c**) females remating (%), (**d**) copulation duration (min), (**e**) spermatophore mass (mg), (**f**) egg fertility (%). Means ± S.E followed by same small letters among different inter-mating intervals (24, 48 and 72 h after initial mating) within a particular regimen of sequential matings of females are not significantly different at *p* < 0.05 level; means ± S.E followed by same capital letters among different sequential mating regimens (of female) within a specific post (first)-mating interval are not significantly different at *p* < 0.05 (2-way ANOVA followed by Tukey’s post hoc test); *n* = 10, where an average reading in a group of 3–4 pairs per cage for calling behavior, copulation duration, spermatophore mass and egg fertility constituted one replicate, whereas 12–15 pairs per cage constituted one replicate for assessing remating success. Percent data were arcsine transformed before ANOVA, but data in graph are back transformations. N♀—unirradiated female, N♂ unirradiated male, T♂_130_—male irradiated with sub-sterilizing dose of 130 Gy, and 130 F_1_♂—F_1_ male progeny derived from sub-sterile male.

## Data Availability

The data presented in this study are available in the article.

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
