# Peer review of "Receptivity and Remating Propensity in Female Spodoptera litura (Fabricius) after Mating with an Irradiated Male or Its F1 Male Progeny"

_insects, 2023, doi:10.3390/insects14070651_

Round 1

Reviewer 1 Report

Good writing but could be improved.

Specific comments:

1. What is the main question addressed by the research?

*The main question addressed by the research is how radiation can reduce the quality of the ejaculate/spermatophore and how this affected the normal female mating function.

2. Do you consider the topic original or relevant in the field? Does it address a specific gap in the field? 

*Yes it does address specific gaps in the field especially on how the female was affected by either the irradiated male and their F1 compared to the normal male. This is not commonly evaluated by researchers

3. What does it add to the subject area compared with other published material?

*It adds to the information on how the irradiated males can affect the functionality of female mating, mating frequency of the female post irradiation. We only usually have mating frequency of normal, non-irradiated males.

4. What specific improvements should the authors consider regarding the methodology? What further controls should be considered?

* I am happy with their methodology which did entails hardwork and did an intuitive approach in how the data gathering was done.

5. Are the conclusions consistent with the evidence and arguments presented and do they address the main question posed?

* Yes, the conclusions were consistent with the results presented and addressed the main hypothesis of the study

English is comprehensive but could still be improved.

Author Response

Response to Reviewer 1 : Comments and Suggestions for Authors

Remarks: Some improvement if possible is stated in various sections , such as Introduction, cited references, Materials & methods, Results & Discussion.

Response: The authors express sincere thanks for the remarks of the Reviewer 1. The manuscript was revisited and relevant additions/changes have been made to make the manuscript more crisp and meaningful.

Reviewer 2 Report

Unfortunately, similar research has already been done, and you must clarify the originality of your own research while citing his work.

Angmo, Nilza, et al. "Effect of male irradiation and its mating status on the remating propensity, insemination quality and reproductive behaviour of male Spodoptera litura (F.)." Indian Journal of Entomology (2023): 95-102.

https://doi.org/10.55446/IJE.2022.693

Commenting on the manuscript itself, the references are inadequate (wrong citations, not cited where they should be), and although there are many papers on the reproductive behavior of the S. litura, the necessary ones are not cited. It would be better to cite well and state where the study stands since there are many papers on this material S. litura rather than generalities in the intro.

non

Author Response

Response to Reviewer 2 : Comments and Suggestions for Authors

Remarks: The improvement is stated in various sections, such as Introduction, Cited references, Materials & methods, Results & Discussion. Commenting on the manuscript itself, the references are inadequate (wrong citations, not cited where they should be), and although there are many papers on the reproductive behavior of the S. litura, the necessary ones are not cited. It would be better to cite well and state where the study stands since there are many papers on this material S. litura rather than generalities in the intro.

Response: The authors sincerely acknowledge the remarks of the Reviewer 2, who was critical enough to improve upon every section of the manuscript.  The manuscript was properly revisited and relevant additions/changes have been made to make the manuscript more explicit and meaningful. Some important references have been added in the text. Various queries have been attended as follows:

Query: Unfortunately, similar research has already been done, and you must clarify the originality of your own research  while citing his work.

Ref: Angmo, N.,  Vimal,  N., Sengupta, M. and Seth, R.K. 2022. Effect of male irradiation and its mating status on the remating propensity, insemination quality and reproductive behaviour of male noctuid pest, Spodoptera litura (F.). Ind. J. Ent. DoI. No.: 10.55446/IJE.2022.693

Response: Indeed it is unfortunate that the Reviewer has misapprehended the situation inadvertently. This is the research work of our unit only, and the objectives are distinctly different in these two papers. For instance, in Angmo et al. 2023 the focus was on the  impact  of male irradiation and its mating status on the remating propensity, insemination quality and reproductive behaviour of male noctuid pest, Spodoptera litura; whereas in the present manuscript, the focus was on the receptivity and remating propensity in female Spodoptera  which had been initially mated with irradiated /or F1 male. Hence, in the present manuscript, main emphasis was laid on the behaviour of female moth in response to her initial mating with irradiated or F1 males.

We have been working on the using nuclear technology in Lepidopteran pest management for the last three decades. The success /efficiency of this radiogenetic technique (F1 Sterility technique) would be influenced by multiple mating nature of this moth, Spodoptera litura.  Therefore, it was intended to study the  reproductive behaviour and viability of irradiated male moths as well as normal (unirradiated) female moths in response to their successive matings.

Incidentally, in our earlier & recently  published paper (Angmo et al 2023) the effect of male irradiation and its mating status was stated on the remating propensity, insemination quality and reproductive behaviour of male noctuid pest with normal females. Whereas in the present manuscript, the focus was laid on the receptivity and remating propensity in female Spodoptera  which was  initially (first time) mated with irradiated male or their F1 male progeny.

Despite the fact, that these two studies looked at some of the same factors and biofeatures, they were conducted independently under different experimental regimens. For instance, Angmo et al. (2023) evaluated the impact of substerilizing and fully sterilizing radiation on mating and reproductive behaviour of male moths; while in the present study, the effect of  mating of substerilized male moths and their F1 progeny was evaluated on the female receptivity and eventual reproductive output.

____________

Reviewer 3 Report

The manuscript titled " Receptivity and remating propensity in female Spodoptera litura (Fabricius) after mating with an irradiated male or its F1 male progeny " is clearly written and relevant information about the factors that affect the male moth irradiation, insemination quality, and post-mating intervals on the remating behaviour when are used in IS insect programmes.

The materials and methods are well-written, and the experimental unit and the design are well-defined.

Some recommendations have been made about the pdf, especially in the discussion in which it is necessary to point out some concepts.

Line 79. Delete “Often”, because in this case the explication is focused on mating and specifically the male,

Line 96. Through the haemolymph, The idea is not clear, please rewrite.

Line 120. “and precautionary measures were taken to avoid microbial infection in the culture” Delete.

Lines 441-450. Move to the introduction, because is not a discussion.

Lines 451-453. This paragraph is really necessary because describes the objective, but not the discussion.

Lines 455-459. The authors cited Seth 458 & Sharma, but no discussion.

Lines 485-486. Explain. The size of the spermatophore is too important

Author Response

Response to Reviewer 3 (Comments and Suggestions for Authors)

Remarks: The reviewer has expressed his satisfaction and approved the various sections, such as Introduction, Cited references, Materials & methods, Results & Discussion. The manuscript titled, “Receptivity and remating propensity in female Spodoptera litura (Fabricius) after mating with an irradiated male or its F1 male progeny” is clearly written and relevant information about the factors that affect the male moth irradiation, insemination quality and post mating intervals on the remating behaviour when are used in IS insect programme.

The materials and methods are well-written, and the experimental unit and the design are well-defined.

Response: The authors express sincere gratitude to the Reviewer for his satisfactory response regarding this manuscript. The queries posed by the Reviewer have been attended properly as follows:

Query: Some recommendations have been made about the pdf, especially in the discussion in which it is necessary to point out some concepts.

Query (i) Line 79. Delete “Often”, because in this case the explication is focused on mating and specifically the male,

Response: Yes, we agree. The deletion was made accordingly.

Query (ii) Line 96. Through the haemolymph, The idea is not clear, please rewrite.

Response: We have rewritten it in an explicit manner.

Query (iii) Line 120. “and precautionary measures were taken to avoid microbial infection in the culture” Delete.

Response: This information has been deleted.

Query (iv) Lines 441-450. Move to the introduction, because is not a discussion.

Response: We agree that this could be a part of Introduction (wherein the concerned information is given in detail),  but here it has been retained to introduce the aim of the study before the results were discussed.

Query (v) Lines 451-453. This paragraph is really necessary because describes the objective, but not the discussion.

Response: Yes we agree that this describes the objective, but this information is retained to indicate the importance of the study before connecting it to discussion that follows.

Query (vi) Lines 455-459. The authors cited Seth  & Sharma, but no discussion.

Response: We have included this reference to corroborate the earlier results.

Query (vii) Lines 485-486. Explain. The size of the spermatophore is too important

Response: The correlation of spermatophore size with its insemination quantity/quality has been elaborated in the text.

Round 2

Reviewer 2 Report

The authors don't seem to understand much of what I have pointed out. You would know, since you yourself are the author of the paper I pointed out. Much of the data is duplicated in the two papers, correct? Much of the data on (N female x N male) and (N female x Tmale130). I understand that the purpose is a bit different, but this is almost a double use of data, just doing the same experiment and looking at it differently.

Since the base is your prior study, you must clearly show how this study is different based on it. Since this paper is not a review paper on the reproductive behavior of Spodoptera litura, you should write a more compact paper based on your paper. There are far too many references cited than necessary.

Extensive editing of English language required

Author Response

Response to Reviewer 2 : Comments and Suggestions for Authors

Remarks/Query: The improvement is referred in various sections of the manuscript.  The authors don't seem to understand much of what I have pointed out. You would know, since you yourself are the author of the paper I pointed out. Much of the data is duplicated in the two papers, correct? Much of the data on (N female x N male) and (N female x Tmale130). I understand that the purpose is a bit different, but this is almost a double use of data, just doing the same experiment and looking at it differently.

Since the base is your prior study, you must clearly show how this study is different based on it. Since this paper is not a review paper on the reproductive behavior of Spodoptera litura, you should write a more compact paper based on your paper. There are far too many references cited than necessary.

Response:

The authors appreciate the keen interest of the Reviewer-2 to make the manuscript more explicit. We would wish to put some observations to understand the genuinity and  relevance of this manuscript, as follows:

Basically, we have been trying to understand the mating behaviour  of substerilized male moths (Spodoptera litura) irradiated with 130Gy (proposed dose in F1 Sterility technique) and its F1 progeny, with normal (unirradiated, wild) female moths. 

There are some special characteristics in Lepidopteran group of insects, such as multiple mating (in both sexes) and sperm dichotomy along with sperm precedence that may be crucial factors in the operational success of radio-genetic ‘Inherited Sterility Technique(IS).

In parent cross, T-male x N-female, the substerilized males would be mating with normal female (unirradiated), but mating behaviour of substerilized male moth and ensuing reproductive success/inviability would be influenced by the level of ionizing radiation administered, and the remating propensity of male moths with normal female. Similar mating dynamics would be exercised by F1 male moths (derived treated male parents)

In case of male, the effect of radiation would be on the insemination quality, mating behaviour and remating propensity vis-a-vis different inter-mating intervals of treated male moths with normal females, that would decide the reproductive success.

Whereas in case of female (normal) moths, its(female’s) premating behaviour, remating propensity vis-à-vis various inter-mating intervals, and reproductive output would be influenced by irradiation stress on its first mate (treated male moth)  or its inherited effect on F1 male progeny.

The goal of the current investigation was to comprehend how the female receptivity affected the effectiveness of the F1 sterility strategy. In the present study, the increased female’s receptivity after its initial mating with F1 male progeny was observed. To further understand the reason of increased female’ receptivity, various parameters necessary to  understand the mating behaviour  were evaluated and therefore, it was necessary to revisit few such bio-parameters with regard to 130 Gy irradiated male so as to obtain a comparative study with respect to its F1 male progeny.

These points are very vital to understand the mating dynamics of treated moths with wild normal females, and model the logistic operation of  such radio-genetic technique.

The  mating & remating behaviour of treated male moths with normal female moths was focused in our first manuscript(Angmo et al . 2023); whereas the mating and remating dynamics of normal female behaviour vis-à-vis irradiated(substerilized) male moths or its F1 male progeny was evaluated in the present study.

We agree  that some of the experiments (for instance copulation duration, mating success, sperm count seemed to be similar, but they were required to be repeated to get the focused information. These sensitive experiments (including irradiated males/F1 males in relation to normal female moths)  were supposed to be done in different regimens for obtaining different information; wherein some experimental observations for ‘control’ need to be repeated for comparison; because ethically we are not supposed to use control/comparison groups from the different regimen experiments conducted at different timings. The reviewer rightly mentioned that the purpose of the recently published  manuscript (focusing on male behaviour) was different from the present manuscript (focusing on female behaviour) in order to obtain  different information about the mating dynamics and reproductive output. Ultimately, the behaviour of both treated male/F1 male moths vis-a-vis wild normal female moths attribute towards overall success of Inherited sterility technique.

Basically, the this manuscript is  complementary to our earlier manuscript (mentioned by the Reviewer.

These observations might be very crucial/vital to decide and optimize the operational logistics of the employment of radio-genetic technique

We have revisited the manuscript again to make some relevant changes to make the manuscript more explicit. Certain references were added in the last revision, based on the suggestions of the Reviewer 2 only; and now on revisiting the text, we feel such references probably should be retained for the meaningful message.